# A Cohort Study Comparing Women with Autism Spectrum Disorder with and without Generalized Joint Hypermobility

**DOI:** 10.3390/bs8030035

**Published:** 2018-03-17

**Authors:** Emily L. Casanova, Julia L. Sharp, Stephen M. Edelson, Desmond P. Kelly, Manuel F. Casanova

**Affiliations:** 1Department of Biomedical Sciences, University of South Carolina School of Medicine Greenville, Greenville, SC 29605, USA; mcasanova@ghs.org; 2Department of Pediatrics, Greenville Health System Children’s Hospital, Greenville, SC 29605, USA; dkelly@ghs.org; 3Department of Statistics, Colorado State University, Fort Collins, CO 80523, USA; julia.sharp@colostate.edu; 4Autism Research Institute (ARI), San Diego, CA 92116, USA; director@autism.com

**Keywords:** connective tissue diseases, autoimmunity, mast cells, immunity, humoral, endocrine system diseases, neurodevelopmental disorders

## Abstract

Reports suggest comorbidity between autism spectrum disorder (ASD) and the connective tissue disorder, Ehlers-Danlos syndrome (EDS). People with EDS and the broader spectrum of Generalized Joint Hypermobility (GJH) often present with immune- and endocrine-mediated conditions. Meanwhile, immune/endocrine dysregulation is a popular theme in autism research. We surveyed a group of ASD women with/without GJH to determine differences in immune/endocrine exophenotypes. ASD women 25 years or older were invited to participate in an online survey. Respondents completed a questionnaire concerning diagnoses, immune/endocrine symptom history, experiences with pain, and seizure history. ASD women with GJH (ASD/GJH) reported more immune- and endocrine-mediated conditions than their non-GJH counterparts (*p* = 0.001). Autoimmune conditions were especially prominent in the ASD/GJH group (*p* = 0.027). Presence of immune-mediated symptoms often co-occurred with one another (*p* < 0.001–0.020), as did endocrine-mediated symptoms (*p* < 0.001–0.045), irrespective of the group. Finally, the numbers of immune- and endocrine-mediated symptoms shared a strong inter-relationship (*p* < 0.001), suggesting potential system crosstalk. While our results cannot estimate comorbidity, they reinforce concepts of an etiological relationship between ASD and GJH. Meanwhile, women with ASD/GJH have complex immune/endocrine exophenotypes compared to their non-GJH counterparts. Further, we discuss how connective tissue regulates the immune system and how the immune/endocrine systems in turn may modulate collagen synthesis, potentially leading to higher rates of GJH in this subpopulation.

## 1. Introduction

Ehlers-Danlos syndrome (EDS) is a group of phenotypically-related disorders subtyped according to variations in underlying genetic pathology, primary symptom severity, and secondary symptom associations. All of these conditions are typified by deficits in collagen production and maintenance, leading to structural changes within the connective tissues of the body. These changes are most evident within the joints and skin, although many other systems can be affected.

Generalized joint hypermobility (GJH) is a major feature of Hypermobile EDS (hEDS) and other connective tissue disorders. In addition, GJH can either be benign or associated with significant musculoskeletal impairment; the latter of which is often affected by an individual’s age, leading to changes in diagnosis over time. According to newer nosology [1], when GJH occurs in conjunction with significant impairment and other criteria for hEDS are not met, it is diagnosed as “Generalized Hypermobility Spectrum Disorder (G-HSD).” Studies have shown that hEDS and GJH often co-segregate within families, indicating linked etiologies in some cases [reviewed in 1].

Neuropsychiatric manifestations are common secondary symptoms in EDS/GJH. In particular, anxiety and mood disorders are prominent and probably the best studied to date [2,3]. However, a thorough review of the literature by Baeza-Velasco et al. [4] suggests significant links between EDS and autism, as well as other neurodevelopmental and psychiatric conditions such as attention-deficit hyperactivity disorder (ADHD), schizophrenia, eating disorders, personality disorders, and even substance abuse. Interestingly, work by Shetreat et al. [5] and Eccles et al. [6] indicates that joint hypermobility is significantly more common in children and adults with autism than age- and gender-matched controls, suggesting etiological links between some cases of autism and connective tissue disorders.

### Immune & Endocrine Dysregulation in ASD & GJH

Immune and neuroendocrine crosstalk is a well-established phenomenon. These systems are linked via two primary pathways through which that crosstalk is achieved: (1) the sympathoadrenal system and (2) the hypothalamo-pituitary-adrenal axis [7]. The immune system can also have a direct effect on oogenesis through the presence of innate and adaptive immune cells located within the ovarian germ cell pool, which release morphoregulatory signals that stimulate or suppress ovulation [8].

Immune dysregulation has been a popular area of study in autism research, whose foci center around topics of maternal immune activation (MIA), prevalence of autoimmunity, and other aspects of general immune dysfunction [9]. In regards to the latter, Careaga et al. [10] have identified two non-overlapping Th1- and Th2-skewed endophenotypes that are especially prominent in children with ASD.

Hormonal exophenotypes, in contrast, have been less well-studied in ASD. One study by Ingudomnokul et al. [11] found that high-functioning women on the autism spectrum and their mothers reported high rates of endocrine disorders. However, most endocrine research to date has focused on maternal disorders with an emphasis on etiological risk factors, such as diabetes, hirsutism, and polycystic ovary syndrome (PCOS) [12,13,14].

High rates of immune- and endocrine-mediated disorders have also been reported in EDS, though they are currently viewed as secondary symptoms to what are traditionally seen as “collagen disorders” [15,16,17]. While it has previously been difficult to explain links between immune and collagen dysfunction, research into the connective tissue disorders, Marfan and Loeys-Dietz Syndromes, which share features of overlap with EDS, may help to guide future EDS research.

In order to study the frequency and relationship of immune and endocrine exophenotypes in adult women with ASD, with or without GJH, we have utilized self-reports covering a range of clinical symptoms, including features of chronic allergies, autoimmunity, irritable bowel syndrome (IBS)/gastrointestinal (GI) dysfunction, and menstrual irregularities.

## 2. Methods

### 2.1. Study Population

The vast majority (94%) of respondents were affected persons themselves, rather than family members responding for adult wards. As such, it is assumed that the majority of our study population was composed of women with an IQ > 70 due to their abilities to answer a series of complex questions about general health.

Our study group was composed of two English-speaking subpopulations: (1) women 25 years and older with a diagnosis of ASD (referred to here as simply “ASD”) (*N* = 85); and (2) women 25 years or older with dual ASD and EDS, G-HSD, or Joint Hypermobility Syndrome (JHS) diagnoses (referred to here as “ASD/GJH”) (*N* = 20) (Figure 1). Individuals who were male, were under the age of 25, or did not have a diagnosis of ASD were excluded. In the ASD group, further exclusionary criteria were applied: (a) An individual’s responses were removed from the data pool if she suspected the presence of GJH but was currently undiagnosed, and/or (b) reported double-jointedness across two or more types of joints [18]. The majority of women reporting EDS diagnosis had hEDS, although a small minority reported diagnosis of Classical EDS. (See Table 1 for descriptions of terms and definitions.)

Our groups were sex-matched and did not differ significantly by age (*t* = −0.327, *df* = 28.451, *p* = 0.7459). Full data are presented in Appendix A. All data were complete, with the exception of two respondents’ answers on the topic of “Other Chronic Pain”.

Due to the biased manner in which respondents were recruited [i.e., specifically targeting both ASD and ASD/GJH subgroups via respective web fora (see Section 2.2 under Methods)], we are unable to estimate the prevalence of GJH in the female ASD population. However, this method allowed us to collect a larger pool of ASD/GJH respondents, which might otherwise be underrepresented. In doing so, we are able to study group differences more easily.

### 2.2. Survey

This study was approved by the Institutional Review Board (IRB) of the Greenville Health System (GHS) (ID: Pro00061122). The survey utilized in this study was designed by our research group based in part on previous informal survey studies performed by the Autism Research Institute (ARI). These questions were further adapted and expanded according to an additional literature search of relevant clinical symptomology for our topics of interest. (See Appendix A for full survey.)

The survey was built on and hosted by the website, SurveyGizmo.com, and was advertised via the ARI newsletter; the ASD forum, Wrong Planet; and a variety of FaceBook ASD- and EDS-specific webcommunities, such as the Autism Women’s Network (AWN), the Autism Spectrum Women’s Group, AutismTalk, the Ehlers-Danlos Support Group, and Ehlers-Danlos Worldwide. The administrative teams of all participating web communities were informed that the survey was IRB-approved and were given access to the survey and, when requested, a copy of the IRB protocol prior to approval. Following administrator approval, either ELC posted the survey announcement or administrators posted it themselves. The survey weblink (www.autismwomensstudy.com) led potential respondents to a description of the purpose and expectations of the study, potential risks and benefits, investigator contact information, and a waiver of consent. The survey was open and participants were actively recruited for approximately three months.

Survey questions focused on topics concerning ASD and EDS/GJH diagnoses; symptoms involving the immune and endocrine systems; chronic pain; GI dysfunction such as IBS; seizures; and limited aspects of medication history (hormone treatment, antiseizure medications). Additional topics were covered but were not used for the current study.

Questions on immune symptomology included items concerning hospitalization history; respiratory disorders like asthma, allergies, sinusitis/rhinitis, and reactions to medications or environmental chemicals; and autoimmunity [19,20,21]. Hormone-mediated symptoms included items such as chronic irregularities in menstruation and associated pain syndromes; PCOS; and other clinical symptoms indicative of the metabolic syndrome, such as type 2 diabetes/insulin resistance, hypertension, and high cholesterol [11,22]. The data of respondents who agreed to participate but failed to complete the survey were discarded.

### 2.3. Statistical Analyses

When assessing group differences for quantitative variables (e.g., age, immune- and endocrine-mediated symptoms), Welch two-sample *t*-tests were conducted unless the distributions were heavily skewed, in which case the Wilcoxon rank sum test with continuity correction was used. Two sample tests of proportions were used to compare groups for binary categorical data (e.g., presence/absence of diabetes, infertility, etc.). Fisher’s Exact Test was used in cases of small sample sizes. Where appropriate, a false discovery rate adjustment was used to account for multiple comparisons. A significance level of 0.05 was used for all analyses.

## 3. Results

### 3.1. Immune-Mediated Disorders

Although women with ASD and ASD/GJH did not differ in the presence of one or more immune-mediated symptoms (*χ^2^* = 1.162, *p* = 0.281), ASD/GJH women were, however, more likely to report multiple symptoms (*t* = −3.860, *df* = 30.981, *p* = 0.001), an effect that differed by age (*W* = 534.5–563, *p* = 0.009–0.017) (Figure 2A). Women with ASD and ASD/GJH also reported similar proportions of specific immune exophenotypes (*χ^2^* = 0.788–4.744, *p* = 0.137–0.375), with an overall trend towards higher proportions in the ASD/GJH group. However, one exception concerned autoimmune disorders: while 13% of the ASD group had an autoimmune disorder, 45% of women with ASD/GJH reported the same (*χ^2^* = 8.813, *p* = 0.027) (Table 2).

The presence of most immune exophenotypes exhibited a significant association with one another, suggesting that a similar etiological background underlies many of these symptoms. Allergies, rhinitis, sinusitis, asthma, ear infections, reaction to medications, and reaction to environmental chemicals all seemed to share a strong interrelationship (*p* < 0.001–0.020). Meanwhile, autoimmunity was significantly associated with ear infections (*CI* = 1.509–18.898, *p* = 0.014) and showed a trend towards significance with asthma (*CI* = 1.003–9.582, *p* = 0.059) (Figure 2B).

Interestingly, the proportions of IBS/GI dysfunction did not differ significantly between groups, though the ASD/GJH group reported modestly higher rates (*χ^2^* = 0.648, *p* = 0.946). In spite of IBS’ links with immunity, it did not share a significant relationship with immune exophenotypes in general (*CI* = 0.282–7.60, *p =* 0.717), although our study may have been too underpowered to glean an effect [31]. Yet in spite of modest numbers, IBS/GI dysfunction was significantly linked with hormonal exophenotypes: individuals with IBS/GI dysfunction had more hormone-mediated symptoms on average than those without (*W* = 811, *p* = 0.002).

In spite of the previously reported relationship between hormones and seizure propensity, the presence of complex hormonal exophenotypes was not associated with epilepsy in our cohort (*W* = 274, *unadj. p* = 0.3742) [32]. However, despite the small number of women reporting epilepsy (*N* = 7), epilepsy shared a modest positive relationship with the number of immune-mediated disorders irrespective of the group (*W* = 166, *unadj. p* = 0.022). While the average number of immune-mediated disorders reported across the entire cohort was approximately 3.6 (*SD* = 2.45), women with epilepsy averaged approximately 5.7 (*SD* = 2.43).

Finally, joint pain was reported in all cases of ASD/GJH compared to 29% in the ASD group (*χ^2^* = 30.122, *p* < 0.001). Meanwhile, differences in joint pain were not accounted for by age (*W* = 559.5, *p* = 0.101) or obesity (*χ^2^*= 0, *p* = 1.000). Other types of chronic pain were also reported more often in ASD/GJH (75% vs. 31%) (*χ^2^* = 11.072, *p* < 0.001), including conditions such as fibromyalgia [33].

### 3.2. Hormone Disorders

Though the ASD/GJH and ASD groups did not differ in the presence of one or more hormone-mediated disorders (*χ^2^* = 0.728, *p* = 0.394), ASD/GJH women reported significantly more symptoms than their non-GJH counterparts (*W* = 434, *p* = 0.001) (Figure 2C). On a symptom-by-symptom basis, ASD/GJH women reported higher rates of endometriosis (*χ^2^*= 9.265, *p* = 0.018), dysmenorrhea (*χ^2^*= 19.599, *p* < 0.001), and severe teen acne (*χ^2^*= 7.817, *p* = 0.026) (Table 3). Dysmenorrhea, in particular, was reported three times more often (85% vs. 28%) in ASD/GJH compared to ASD (*χ^2^*= 19.60, *p* < 0.001), a frequency similar to that reported in previous EDS research (see Table 3). In its extreme form, dysmenorrhea is typically associated with endometriosis and both share links with immune dysfunction in the general population [34,35].

We found no significant interaction between group and birth control/hormone treatment in relation to the average number of hormone-mediated symptoms, indicating that such treatment is an unlikely group confound in this study. There was, however, a significant relationship between the number of hormone-mediated symptoms reported and whether an individual was receiving some form of hormonal treatment (*t*(*101*) = 2.75, *p* = 0.004). While we cannot rule out a potential confound, instead, we conclude that this is likely a reflection of the severity of endocrine disorders in our cohort and their prescribed treatments [51].

Like immune symptoms, individual hormone-mediated symptoms were often associated with one another (Figure 2D). PCOS shared links with other symptoms, including diabetes, adult acne, irregular menses, and hirsutism, all of which are either diagnostic of or commonly reported in PCOS (*p* = 0.005–0.045). In contrast, infertility, overweight/obesity, amenorrhea, hypertension, and high low-density lipoprotein (LDL) cholesterol did not associate with PCOS in our groups (*p* = 0.073–0.809) [52]. There was, however, a trend towards significance between PCOS and infertility, suggesting our data may have been underpowered, requiring a larger pool of respondents in the future (*OR 95% CI* = 1.20–37.800, *p* = 0.073). Endometriosis and dysmenorrhea were also associated with one another (*CI* = 2.229–785.323, *p* = 0.013).

### 3.3. The Relationship between Immune- & Hormone-Mediated Symptoms

There was no significant association between the general presence of immune- and endocrine-mediated disorders across our cohort (*OR 95% CI* = 0.538, 21.119, *p* = 0.098). However, the number of immune-mediated symptoms per individual greatly predicted the number of hormone-mediated symptoms (*Spearman’s rho* = 0.35, *p* < 0.001). This suggests that the complexity and severity of immune- and endocrine-mediated disorders share a strong positive relationship with one another in autism and potentially within the general population, e.g., [53].

## 4. Discussion

The present study attempts to address phenotypic differences between ASD women with and without GJH. This research supports a growing body of literature indicating that immune-mediated disorders are a common comorbid feature in hEDS and GJH. In addition, we have also shown that this dysfunction may be paired with endocrine dysregulation, leading to complex immune and hormonal exophenotypes, such as autoimmune disorders, allergic rhinitis, asthma, endometriosis, and dysmenorrhea. While we have not addressed autism and GJH comorbidity rates in this study, their co-occurrence in the adult ASD female population suggests links between the dysfunction of connective tissue and the immune and endocrine systems in this subpopulation.

As discussed, the immune system has been a popular area of investigation in autism research. However, reports of clinical manifestations in the child population seem to vary [54,55,56,57]. Some clinical manifestations arise during or progress in severity with the advent of puberty, highlighting the role the endocrine system plays in immune function, e.g., [58]. In addition, women are more frequent targets of such dysfunction, suggesting that studying immune dysregulation in prepubertal individuals with autism, while also ignoring gender confounds, dramatically underrepresents the frequency of clinical symptoms in the autism population [19]. For these reasons, we limited our study population to women aged 25 years or older on the autism spectrum.

### 4.1. Immune-Mediated Disorders in Association with Connective Tissue Disorders

Loeys-Dietz Syndrome (LDS) is a connective tissue disorder caused by mutations directly targeting the TGF-β pathway and is characterized primarily by enlargement of the aorta. People with LDS have high rates of immune-mediated disorders such as respiratory and food allergies and occasionally present with Hyper-IgE Syndrome, a type of primary immunodeficiency [59]. In addition, they also share many of the same dysmorphic features as those seen in the connective tissue disorder, Marfan Syndrome (MFS) [60].

Although MFS is associated with mutations in the *Fibrillin-1* (*FBN1*) gene whose protein product is a component of the extracellular matrix (ECM), *FBN1* mutations lead to marked TGF-β dysregulation [61,62,63]. Fibrillin appears to control the activity of TGF-β by acting as a structural platform for the Latent TGF-β Binding Protein (LTBP) that sequesters and inactivates TGF-β, acting as a reserve pool for rapid injury response [64]. Given its role as a foundational morphogen, it is believed this overlap in TGF-β pathway dysregulation leads to the overlapping features of MFS and LDS [65].

Like LDS, some individuals with hEDS present with a Marfanoid (Marfan-like) habitus [66]. However, unlike MFS that results from dysfunctional fibrillin, EDS is typically linked with dysfunction of the ECM protein, collagen. Marfan and Marfanoid features in all three of these disorders suggest considerable overlap and interaction between the ECM and the TGF-β pathway. In addition, TGF-β serves as a link between the ECM and immune system disruption as it is a key immunomodulator, implicated not only within the joints in these connective tissue disorders, but also in other organ systems such as the lungs [65]. Interestingly, several studies have consistently found lower TGF-β1 levels in autism, which according to Ashwood et al. [67], may help explain some of the immune dysregulation in the condition [67,68]. For these reasons, the TGF-β pathway and upstream networks may be prime areas of study for future work into the overlapping etiologies of both connective tissue disorders and autism.

### 4.2. The Effects of Estrogen on Collagen Production & the Immune System

Similar to certain immune disorders like autoimmunity, GJH and hEDS preferentially target women for reasons not well understood [69]. One possibility may stem from sex differences in muscle mass, in which stronger muscles help to counteract joint laxity and ensuant pain [16]. For this reason, one of the foci of physical therapy in the treatment of GJH/hEDS centers around improved muscle strength surrounding problem joints [70]. However, female-specific effects may result not only from low testosterone levels, but also estrogen metabolites that either suppress collagen production directly, particularly within the skin, or result in a more rapid turnover of collagen within tendons and ligaments [71,72,73].

Estrogen is also a major immunomodulator. It is capable of driving activation of the Th2 branch of the immune system, boosting humoral immunity and the ability of the body to target parasites and other extracellular infections. Estrogen also stimulates mast cell degranulation, prompting a release of chemicals such as histamines, TNF-α, various amines, chymase, and tryptase [74,75]. Mast cell activation, in turn, may drive both Th1/Th2 immune responses depending on the invading pathogen, the target tissue, and other variable factors [76].

Interestingly, estrogen also increases the synthesis of TGF-β within numerous cell types, the latter of which is itself a key morphogen and immunodulator. In addition, estrogen further interacts with the TGF-β pathway by forming a complex with Smad 3/4, redirecting TGF-β target genes. Finally, TGF-β and estrogen are able to interact at the level of various Ras complexes, by which TGF-β enhances estrogenic action [77]. All of these data together suggest significant interaction of estrogen with various networks implicated in connective tissue disorders and their secondary symptoms.

### 4.3. Autism & Generalized Joint Hypermobility

Results of this study indicate that the ASD/GJH phenotype in women is characterized not only by classic symptoms of EDS/G-HSD such as generalized hypermobility and chronic pain, but that immune and endocrine system involvements may be extensive. In addition, phenotypic expression of this immune disorder is mediated by the endocrine system and the ongoing presentation of symptoms throughout life are guided by immune-endocrine crosstalk.

In support of this, all 20 ASD/GJH women in our study group reported ≥ 2 immune-mediated symptoms, with an average reporting of 5.3 symptoms per person compared to 3.2 in the ASD group. Likewise, 90% of ASD/GJH women reported ≥ 2 hormone-mediated symptoms, with an average of 5.1, compared to 2.7 in ASD. Therefore, the vast majority of ASD/GJH women in this study reported multiple immune- and endocrine-mediated symptoms, the extent of which appears to vary with one another.

Mast Cell Activation Syndrome (MCAS), a newly recognized diagnostic entity with growing clinical significance, may be relevant to immune exophenotypes reported by our participants [15]. While the traditional slew of MCAS impairments include analphylaxis, syncope, flushing, urticaria, and GI distress (e.g., diarrhea, nausea, vomiting), continued study of this condition reveals a broader spectrum of physical ailments relative to the locations of mast cells involved, the extent of stimulation, and the specific mediators released.

Although MCAS can mimic many localized diseases, its defining feature is chronic mast cell activation across two or more organ systems, which is reminiscent of the complex combination of respiratory, connective tissue, and GI symptoms reported by some of our participants [78,79]. Interestingly, MCAS is also a common comorbid feature of EDS and postural orthostatic syndrome (POTS), reinforcing this emerging pattern [15,80]. Current prevalence rates of this newly recognized entity (14–17%) also suggest it is far more common in the general population than originally believed [78].

While GJH can occur without complications, many cases involve extensive inflammation at the affected joints, suggesting a potential immune component in the disorder as is seen in TGF-β pathway involvement in LDS and MFS. As Afrin [78] suggests in reference to the MCAS-/hEDS relationship:
*… chronic aberrant elaboration of a particular set of mediators (drawn from amongst the mast cell’s repertoire of more than 200 such molecular signals) not only [influences] virtually every other system and organ in the body but also [influences] connective tissue development to yield the “hyperextensible” phenotype long associated with EDS Type III [(hypermobile type)]*.(p. 138)

### 4.4. The Etiology of Autism

While this study cannot address rates of ASD and GJH co-occurrence because of the way in which respondents were recruited, the comorbidity itself reinforces etiological links between autism and connective tissue disorders. Both cytokines and hormones play recognized roles in neurogenesis, neuritogenesis, synaptogenesis, and ongoing plasticity [81,82,83,84]. In addition, some researchers have proposed that autoantibodies to brain-specific proteins may also disrupt neurodevelopment, leading to increased autism risk [85]. Finally, endocrine disruption, either via endogenous or exogenous effectors, is likewise a growing area of research into autism’s etiology [12,86]. All of these topics highlight the crosstalk between the immune and endocrine systems and strengthen their combined links to ASD.

### 4.5. Limitations

According to recent changes in nosology, hEDS, the most common of the Ehlers-Danlos Syndromes, lies on a continuum with Hypermobility Spectrum Disorders (HSD), including what was once known as Joint Hypermobility Syndrome (JHS) (see Table 1). Previous studies have shown that hEDS and JHS often co-segregate within families, suggesting that in some cases, JHS/HSD may be a lighter variant of hEDS (reviewed in [1]).

As of last year, the criteria for hEDS have become more stringent, placing greater focus on the additional involvement of tissue systems outside that of the musculoskeletal system, e.g., skin and other organs [1]. It is therefore possible and probable that some individuals in this study who had a previous diagnosis of EDS, Hypermobile Type, no longer reach the cut-off for hEDS and would instead be given a diagnosis of Generalized HSD (G-HSD) were they reassessed.

Due to the nature of online surveys and our inability to reassess participants for appropriate recategorization, it is therefore assumed that the ASD/GJH group in this study contains a mix of individuals who would currently be defined as G-HSD and hEDS. For these reasons, our results may not be fully applicable to hEDS and must therefore be interpreted with caution.

Other limitations of our study concern the reliability of data derived from self-reports, which is vulnerable to reporting bias. In particular, the similarity between rates of clinical presentation in our ASD group and the general population suggests reporting reliability (Table 2 and Table 3). Meanwhile, similarly high rates of immune- and endocrine-mediated disorders in our ASD/GJH group compared to the general HSD/EDS population also support the veracity of their reports [17,33,69].

A related vulnerability of our data hinges on ASD and GJH diagnostic reliability. While the data is dependent upon self-reports, we did however offer respondents the opportunity to specify whether they were professionally diagnosed or suspected a diagnosis. Those who indicated a suspicion of hEDS or some type of HSD were initially included in the first round of analyses as an additional group of interest. However, their data varied too dramatically from the diagnosed group and were not included in the final analysis. Therefore, while the diagnostics are not standardized in this study, those reporting professional diagnoses of ASD or GJH were assumed to be truthful.

Another limitation concerns small sample sizes, particularly of the ASD/GJH group. Given the rarity of EDS (1:5000) and the infrequency of its overlap with ASD (3%), a sample size of 20 could be considered quite large [87,88]. There are unfortunately no current estimates of G-HSD prevalence under the new nosology; however, our results indicate that we have had ample power for this study.

We selectively surveyed ASD women aged 25 years or older to study specific immune and endocrine exophenotypes. However, we cannot generalize our results to the broader autism spectrum, though previous studies indicate that related endo- and exophenotypes exist in ASD males and individuals under the age of 25. Likewise, we cannot generalize our data to the full EDS and GJH spectrums, though previous research supports our findings [17,79,80,89]. Instead, future research is needed to explore a potential clinical spectrum that spans the sexes and the lifespan to determine to what extent our findings apply to the broader autism spectrum and GJH.

Finally, our results suggest there may be a relationship between epilepsy and immune symptomology, which is supported by the recognized roles that cytokines and other immune factors play in epileptogenesis [90]. However, due to small participant numbers, further investigation is necessary to address this potential and is a topic we will be addressing in future studies.

## Figures and Tables

**Figure 1 behavsci-08-00035-f001:**
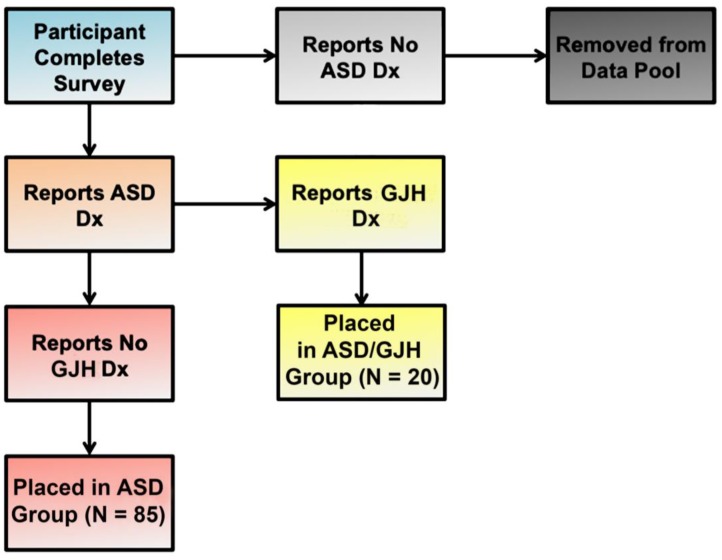
Flow chart illustrating group allocation according to reported diagnoses.

**Figure 2 behavsci-08-00035-f002:**
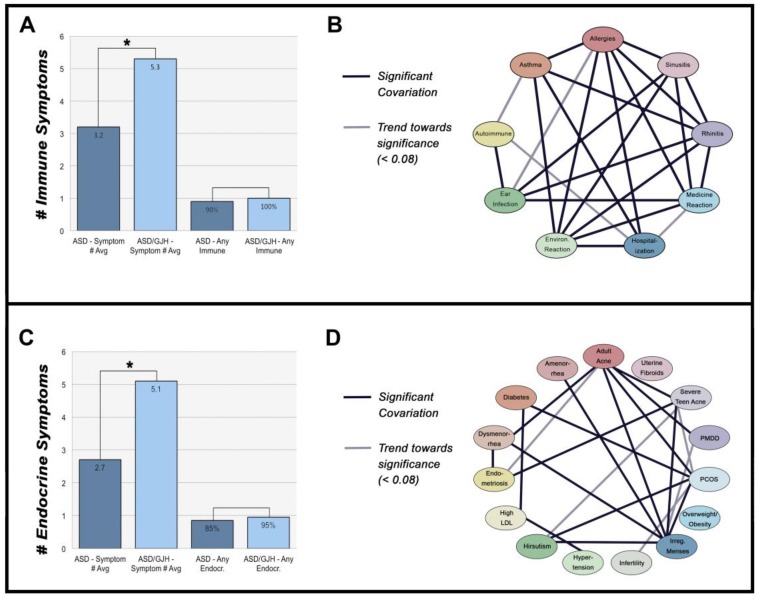
(**A**) Number of immune-mediated symptoms across ASD and ASD/GJH groups. ‘Any immune’ = 1 or more immune symptoms. (**B**) Network of immune-mediated symptoms. (**C**) Number of endocrine-mediates symptoms across ASD and ASD/GJH groups. ‘Any endocr.’ = 1 or more endocrine symptoms. (**D**) Network of endocrine-mediated symptoms.

**Table 1 behavsci-08-00035-t001:** Terms and diagnoses related to the connective tissue disorders discussed in the manuscript.

Generalized Joint Hypermobility-Related Diagnoses	Description
**Hypermobile Ehlers-Danlos Syndrome (hEDS)—Formerly known as EDS, Hypermobile Type, or EDS Type III.**	Generalized joint hypermobility Musculoskeletal involvement (arthralgia, instability) Involvement of other organ systems (skin, Marfanoid features, etc.) No consistently associated gene mutations
**Classical Ehlers-Danlos Syndrome (cEDS)—Also known as EDS Type I.**	Skin hyperpextensibility and atrophic scarring Generalized joint hypermobility Minor features: e.g., easy bruising, skin fragility, hernias, etc. Associated gene mutations: *COL1A1*, *COL5A1*, and *COL5A2*
**Generalized Hypermobility Spectrum Disorder (G-HSD) - Formerly known as “non-benign” JHS.**	Generalized joint hypermobility Musculoskeletal involvement (arthralgia, instability) Other minor criteria associated with hEDS may be present but to a comparatively lesser extent
*** Joint Hypermobility Syndrome (JHS)—Divided into “benign” and “non-benign” forms. Diagnosis now in disuse as of 2017.**	Generalized joint hypermobility Optional: musculoskeletal involvement (arthralgia, instability)
**Hypermobility Spectrum Disorders (HSD)**	**Composed of:** G-HSD (formerly known as “non-benign” JHS) Peripheral HSD (P-HSD) Localized HSD (L-HSD) Historical HSD (H-HSD)
**Asymptomatic Joint Hypermobility**	Asymptomatic Generalized Joint Hypermobility (A-GJH) (formally known as “benign” JHS) Asymptomatic Peripheral Joint Hypermobility (A-PJH) Asymptomatic Localized Joint Hypermobility (A-LJH)
**Marfan Syndrome (MFS)**	Aortic root dilation Ectopia lentis (dislocated lenses of the eye) Minor features: Marfan habitus, generalized joint hypermobility Associated gene mutations: *FBN1*
**Loeys-Dietz Syndrome (LDS)**	Enlargement of the aorta Aneurysms Hypertelorism Bifid uvula or cleft palate Minor features: Marfanoid habitus, immune disorders (allergy, asthma, rhinitis, eczema) Associated gene mutations: *TGFBR1*, *TGFBR2*, *SMAD3*, *TGFB2*, and *TGFB3*

* indicates terminology that is no longer in use as of the recent nosological changes enacted [1].

**Table 2 behavsci-08-00035-t002:** Reported rates of various immune-related symptoms according to group, as well as estimated general prevalence rates.

Immune Symptomology	ASD (*N* = 85)	ASD/EDS (*N* = 20)	General Prevalence
Allergies	45%	60%	30% in adults [23]
Asthma	33%	60%	8.4% in general population [24]
Autoimmunity	13%	45% *	7.6–9.4% in general population [25]
Chronic Ear Infections	40%	65%	83% with ≥1 incidents between 0–3 years of age [26] 11% with ≥1 incidents of all ages [27]
Chronic Rhinitis	38%	60%	8% in adults [28]
Chronic Sinusitis	46%	60%	8% in adults [28]
Severe Reaction to Medications	35%	65%	10–15% of hospitalized patients [29]
Severe Reaction to Environmental Chemicals	39%	65%	13–16% in adults [30]

* indicates a significant difference between groups.

**Table 3 behavsci-08-00035-t003:** Reported rates of various endocrine-related symptoms according to group, as well as estimated general prevalence rates.

Endocrine Symptomology	ASD (*N* = 85)	ASD/EDS (*N* = 20)	General Prevalence
Adult Acne	21%	35%	35% in women ages 30–39 [36]
Amenorrhea	39%	45%	4.6% in women ages 15–44 [37]
Diabetes/Insulin Resistance	6%	10%	7.9% in adults [38]
Dysmenorrhea	28%	85% *	2–29% in adult women [39]
Endometriosis	5%	30% *	4% in women [40]
High LDL Cholesterol	14%	30%	28% in adults [41]
Hirsutism	19%	30%	10% in adult women [42]
Hypertension	14%	20%	29.1% in adults [43]
Infertility	8%	15%	6% [44]
Irregular Menstruation	27%	55%	18.2% in adult women [45]
Overweight/Obesity	36%	45%	70.7% aged 20+ years [46]
Polycystic Ovary Syndrome (PCOS)	8%	25%	7.3% in adult women [47]
Premenstrual Dysphoric Disorder (PMDD)	21%	30%	3–8% of premenopausal women [48]
Severe Teen Acne	14%	45% *	12.1% in males and females aged 17 [49]
Uterine Fibroids	9%	5%	4.5–9.8% in adult women aged 40–49 [50]

* indicates a significant difference between groups.

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
