# Peer review of "A Cohort Study Comparing Women with Autism Spectrum Disorder with and without Generalized Joint Hypermobility"

_behavsci, 2018, doi:10.3390/bs8030035_

Round 1

Reviewer 1 Report

please see the file attached

Author Response

In my opinion the title does not reflect clearly the study. The study compares ASD with and without gJH.

>>> The title has been altered to reflect this suggestion.

Because the groups (ASD+EDS-gJH vs ASD) are established a posteriori, I think that is better to add a first objective which is to explore the frequency of EDS and/or gJH in ASD women. And then to explore differences in immune/endocrine exophenotypes between the formed groups. Accordingly, the results section should start with the description of the frequency of EDS or gJH observed.

>>> Due to the ways in which respondents were recruited, which included advertisement in EDS/HSD-specific groups, we cannot estimate co-occurrence of ASD and GJH in this study. This method allowed us to collect a larger ASD/GJH subgroup, but is not likely an accurate representation of GJH in the larger female ASD population. Therefore, we cannot address GJH frequency. We have addressed this issue now in the text:

“Due to the biased manner in which respondents were recruited, specifically targeting both ASD and ASD/GJH subgroups via respective web for a (see 2.2 Survey under Methods), we are unable to estimate prevalence of GJH in the female ASD population. However, this method allowed us to collect a larger pool of ASD/GJH respondents, which might otherwise be underrepresented. In doing so, we are able to study group differences more easily.”

There are some problems with the structure of the manuscript. It seems rather a draft or the text from an unexperienced writer. For exemple, The introduction lacks of pertinent references. E.g. “According to newer nosology …» but it’s not mentioned what is the new nosology with the correspondant reference, even if this is present in the list of references (66). In addition, I suggest revise primary data and others works concerning psychiatry and ASD suggested links, specially because there few studies and clinical reports so the deserved to be mentioned (e.g. Bulbena et al., TanTam et al, Baeza-Velasco et al., Takei et al, Shetreat-Klein …)

>>> The review section on neuropsychiatric comorbidities has been updated to reflect this body of literature:

“Neuropsychiatric manifestations are common secondary symptoms in EDS/GJH. In particular, anxiety and mood disorders are prominent and probably the best studied to date [2-3]. However, a thorough review of the literature by Baeza-Velasco et al. [4] suggests significant links between EDS and autism, as well as other neurodevelopmental and psychiatric conditions such as attention-deficit hyperactivity disorder (ADHD), schizophrenia, eating disorders, personality disorders, and even substance abuse. Interestingly, work by Shetreat et al. [5] and Eccles et al. [6] indicates that joint hypermobility is significantly more common in children and adults with autism than age- and gender-matched controls, suggesting etiological links between some cases of autism and connective tissue disorders.”

At the end of the introduction the authors exposed the methodology and some results. These should be given in the method and results sections only. At the end of the introduction authors should evoke the objective of the study but not the results.

>>> This has been removed.

The method section usually starts with the description of the participants or population and then the description of instruments (survey), end procedure.

>>> These sections have now been rearranged.

Please define acronyms the first time (e.g. IG, IBS)

>>> These terms were described in the final paragraph of the introduction.

What do you mean with “and limited aspects of medication history »?

>>> We specifically asked regarding the use of hormone treatment and antiseizure medications. This has been specified now in the text.

It suggests to add the corresponding N in the groups of the flowchart (Figure 1).

>>> Corresponding N’s have been added to Figure 1.

There are too many figures and tables, authors should synthetize the information.

>>> Number and complexity of figures have been reduced.

Please clarify “Although this study does not contain a traditional control group representative of the general population, the ASD group serves as a comparison group for ASD/GJH “ - is it a cross-sectional study that actually pretended to be a case-control study? This point can be clarified considering my suggestion of reformulate the objectives.

>>> This section has been removed to reduce confusion.

The first sentences of the discussion don’t correspond to the objective of the study. Maybe authors should start with remembering to the readers the objectives and then organize the discussion according to that.

>>> The objective has been reiterated at the start of the Discussion section.

Reviewer 2 Report

This article describes a web based survey research showing immune/endocrine dysregulation in hEDS and GJH in ASD female patients.

My comments/questions are:

- As this research is a patient-based web survey, what are the exclusion criteria?

- Were women with ASD high functioning? There was a classification in regarding the autism spectra?

- Paragraph 4.1: TGF-b involvement in ASD could be added.

Author Response

As this research is a patient-based web survey, what are the exclusion criteria?

>>> The following has been added to the methods: “Individuals who were male, were under the age of 25, or did not have a diagnosis of ASD were excluded.”

Were women with ASD high functioning? There was a classification in regarding the autism spectra?

>>> We did not have a means to confirm level of functioning, however, the following was stated in the manuscript: “The vast majority (94%) of respondents were affected persons themselves, rather than family members responding for adult wards. As such, It was assumed that the majority of our study population was composed of women with an IQ > 70 due to their abilities to answer a series of complex questions about general health.” We do not know about other levels of functioning, such as daily living habits; only probable levels of cognitive functioning.

Paragraph 4.1: TGF-b involvement in ASD could be added.

>>> Data concerning TGF-beta and ASD has been added to the Discussion.

Round 2

Reviewer 1 Report

My comments have been adequately answered. The current version is more clear and more structured.